# A Prospective Observational Study of Frailty in Geriatric Revitalization Aimed at Community-Dwelling Elderly

**DOI:** 10.3390/jcm13092514

**Published:** 2024-04-25

**Authors:** Almudena Morales-Sánchez, José Ignacio Calvo Arenillas, María José Gutiérrez Palmero, José L. Martín-Conty, Begoña Polonio-López, Luís Alonso Dzul López, Laura Mordillo-Mateos, Juan José Bernal-Jiménez, Rosa Conty-Serrano, Francisca Torres-Falguera, Alfonso Martínez Cano, Carlos Durantez-Fernández

**Affiliations:** 1Emergency Medical Service of Castilla y León (SACYL), 47007 Valladolid, Spain; amorales@saludcastillayleon.es; 2Department of Nursing and Physiotherapy, University of Salamanca, 37007 Salamanca, Spain; calvoreh@usal.es; 3Department of Biomedical and Diagnostic Sciences, University of Salamanca, 37007 Salamanca, Spain; mjgp@usal.es; 4Department of Nursing, Physiotherapy and Occupational Therapy, Faculty of Health Sciences, Universidad de Castilla-La Mancha, 45600 Talavera de la Reina, Spain; begona.polonio@uclm.es (B.P.-L.); laura.mordillo@uclm.es (L.M.-M.); juanjose.bernal@uclm.es (J.J.B.-J.); francisca.torres@uclm.es (F.T.-F.); alfonso.martinez@uclm.es (A.M.C.); 5Technological Innovation Applied to Health Research Group (ITAS), Faculty of Health Sciences, University of Castilla-La Mancha, 45600 Talavera de la Reina, Spain; 6Department of Project Management, Universidad Internacional Iberoamericana (UNINI-MX), Campeche 24560, Mexico; luis.dzul@unini.edu.mx; 7Higher Polytechnic School, Universidad Europea del Atlántico (UNEATLANTICO), 39011 Santander, Spain; 8Faculty of Nursing, University of Castilla-La Mancha, 45071 Toledo, Spain; rosamaria.conty@uclm.es; 9Department of Nursing, Faculty of Nursing, University of Valladolid, 47003 Valladolid, Spain; carlos.durantez@uva.es; 10Nursing Care Research (GICE), University of Valladolid, 47003 Valladolid, Spain

**Keywords:** community dwelling, multicomponent exercise program, older adults, physical frailty, pre-frailty

## Abstract

(1) **Background:** The increasing life expectancy brings an increase in geriatric syndromes, specifically frailty. The literature shows that exercise is a key to preventing, or even reversing, frailty in community-dwelling populations. The main objective is to demonstrate how an intervention based on multicomponent exercise produces an improvement in frailty and pre-frailty in a community-dwelling population. (2) **Methods:** a prospective observational study of a multicomponent exercise program for geriatric revitalization with people aged over 65 holding Barthel Index scores equal to, or beyond, 90. The program was developed over 30 weeks, three times a week, in sessions lasting 45–50 min each. Frailty levels were registered by the Short Physical Performance Battery, FRAIL Questionnaire Screening Tool, and Timed “Up & Go” at the beginning of the program, 30 weeks later (at the end of the program), and following 13 weeks without training; (3) **Results:** 360 participants completed the program; a greater risk of frailty was found before the program started among older women living in urban areas, with a more elevated fat percentage, more baseline pathologies, and wider baseline medication use. Furthermore, heterogeneous results were observed both in training periods and in periods without physical activity. However, they are consistent over time and show improvement after training. They show a good correlation between TUG and SPPB; (4) **Conclusions:** A thirty-week multicomponent exercise program improves frailty and pre-frailty status in a community-dwelling population with no functional decline. Nevertheless, a lack of homogeneity is evident among the various tools used for measuring frailty over training periods and inactivity periods.

## 1. Introduction

An increasing life expectancy and the improvements in healthcare in today’s society have led to a change in the population pyramid. Europe is aging; 21% of the European population was 65 years old or older in 2020, in contrast to 2001’s 16%, which means an increase of 5.5 percent points in almost two decades [1]. The current life expectancy in Europe is 80.1 years (77.2 for men and 82.9 for women) [2]. However, with regards to years of healthy life, the numbers drop to 64.2 years for women and 63.1 years for men. That is, they enjoy good health during 80% of their life expectancy [3]. The situation requires a change to international policies that address social and healthcare needs and are focused on health promotion and prevention for the elderly to allow them healthy aging and quality of life in the years to come [4].

Geriatric syndromes such as frailty, sarcopenia, weight loss, and dementia are very frequent in older people [5]. In systematic reviews and meta-analyses, data show considerable and significant heterogeneity between countries, suggesting the need for more research. But frailty shows a prevalence rate of 12% using physical frailty measures and 24% using a frailty index (FI) [6,7]. Scientists have not reached an agreement on a clear definition of the term “frailty”, but biologically, it is a reversible, dynamic, and multifactorial clinical status [8] characterized by increasing vulnerability of individuals to developing negative healthcare situations (e.g., disability, hospitalization, institutionalization, and death) as a result of exposure to exogenous or endogenous stress factors [9,10]. All of this is causing an increase in the need for long-term care (LTC): home care help, senior residences, and prolonged hospitalization. According to the Organization for Economic Co-operation and Development (OECD) statistics, the average LTC expenditure in Europe as a percentage of the GDP was 1.47%, with expectations of it tripling by 2050 [11].

Among the interventions in frailty, exercise and multimodal interventions (consisting of aerobic endurance, muscle strength, flexibility, stability exercises, and walking) show growing evidence to become a key to preventing, or even reversing, frailty in community-dwelling populations more effectively [12,13]. Nevertheless, no clear recommendations are available on the best exercise routines (number of sessions per week, length, intensity, etc.) [14,15].

This study aims to demonstrate how intervention through a multicomponent exercise program—a geriatric revitalization program—improves frailty and pre-frailty in a community-dwelling population. Additionally, the performance of various frailty measurement tools is analyzed.

## 2. Materials and Methods

### 2.1. Study Design

A prospective observational study was carried out within the Geriatric Revitalization program promoted by the University of Salamanca and developed by the City Hall of Salamanca in various neighboring older people organizations and day-care centers, with a duration of one year (measurements were taken in week 1, 30 weeks later, at the end of the program, and in week 43, after 13 weeks without training).

### 2.2. Intervention

The Geriatric Revitalization Program, initiated in 1994, encompasses a multi-component exercise regimen. Each session incorporates myofascial stretching of major muscle groups, cardio-circulatory conditioning, exercises aimed at enhancing muscle strength and power, as well as activities geared towards improving coordination, flexibility, and agility. The sessions were conducted by physiotherapists, adapting intensity and progression to individual participants, taking into account their varying ages and physical conditions. Participants attended three sessions per week over 30 weeks, with each session lasting approximately 45–50 min. These sessions are conducted in municipal facilities specially adapted for this program. Group sizes are variable, with groups structured to include participants with similar conditions and support needs. The structure of the sessions was: -Stretching and warm-up: a three-minute light cardiovascular and muscle workout.-Joint and muscle exercises: various exercises standing on two feet, sitting, and in decubitus position. For maintaining and increasing joint movement and boosting muscle strength and endurance.-Moving 1: three-minute light jogging, avoiding feelings of fatigue. For activating cardiorespiratory function and increasing functional capability.-Break for hydration: compulsory five-minute pause to drink water.-Agility, coordination, and stability exercises: using different equipment and adapted traditional games.-Moving 2: alternating six-minute light jogging with easy walking and finishing with light jogging again (two minutes for each exercise).-Cooling down and exercises in progression: easy walking, light stretching of the lower limbs combined with deep breathing.-Hydration: onsite hydration and at home.

### 2.3. Participants

The recruitment of the target group was carried out by the researchers between participants in the Geriatric Revitalization program based on predetermined inclusion criteria. Inclusion criteria for participants: age over 65 years, participating in the Geriatric Revitalization program, attending the program’s initial assessment and agreeing to participation in the study by informed consent, and showing Barthel Index scores equal to 90 or higher. Exclusion criteria: missing any of the assessment or clinical evaluation sessions during collection periods or abandoning the Geriatric Revitalization program.

### 2.4. Variables

Prior to intervention, patients’ frailty status was registered following the updated consensus document on frailty prevention in older people made by the Spanish Government Healthcare Department [16]. For that purpose, three validated frailty assessment scales were created:-Short Physical Performance Battery (SPPB) [17] consists of three hierarchical tests (balance test, walking speed test, and standing up from a chair test). Scores for each test up to 4. It helps identify frailty, pre-frailty, and robustness over scores of 0–9, 10–11, and 12, respectively.-FRAIL Questionnaire Screening Tool [18,19] involves five questions (Do you feel tiredness? Can you not go up a staircase to a higher floor? Can you not walk a block? Were you diagnosed with more than five diseases? Have you lost more than 5% of your weight in recent months?). Traditionally, three or more affirmative answers would indicate frailty, and one or two affirmative answers would indicate pre-frailty. The latest evidence points out suspicion of frailty in one or more affirmative answers.-Timed “Up & Go” [20] measures the amount of time a person takes to stand up from a chair, walk three meters at a regular pace, return to the chair, and sit down. Records over 12 s determine a high probability of frailty.

To analyze the effects of exercise on frailty, measurements were also taken 30 weeks later, at the end of the program, and in week 43, after 13 weeks without training. All tests were run by qualified professionals.

In addition, other participants’ demographic and clinical variables were registered at the beginning of the study, including age, sex, place of residence, type of cohabitation, previous falls, time participating in the Geriatric Revitalization program, physical activity (now and five years ago), weight, size, body fat percentage, chronic diseases, and medicines regularly taken. Basic day-to-day activities were assessed as well, according to the Barthel Index. Instrumental activities of daily living were evaluated by applying Lawton Brody’s scale, Yesavage’s Geriatric Depression Scale, and Lobo’s cognitive minitest (MEC). Anthropometric measurements were taken by means of a portable digital balance model PPW3300/01, Omron BF 300^®^ (OMRON, Matsukasa Co. LTD, Kyoto, Japan), and time was recorded using the chronometer Onstart 110 (Decathlon, Villeneuve d’Ascq, France).

### 2.5. Statistical Analysis

The results became visible through the quantitative variables, by mean and standard deviation, including median and interquartile range in the violin plots. The qualitative variables were defined by the relative frequency distribution. Based on the size of the sample, a regular variable distribution was assumed; the Student’s *t*-test and ANOVA (post-hoc Bonferroni) served to compare means in quantitative variables, and the Chi-Square test was applied with regards to qualitative variables. The analysis of changes in time for all tests was performed by McNemar. The Pearson Correlation Coefficient was used to check TUG test time records against other test results (SPPB and FRAIL). IBM SPSS software, Statistics for Windows (version 26, Armok, NY, USA: IBM Corp.), was employed for the statistics, whereas graphs were created with R software (version 4.3.0, Foundation for Statistical Computing, Vienna, Austria). The statistical significance level was *p* < 0.05. The statistical analysis was conducted by investigators separate from those involved in recruitment and intervention. 

### 2.6. Ethical Considerations

This study was approved by the Ethics Committee of the University of Salamanca (reg. no. 307). In conformity with the provisions of the Declaration of Helsinki, every participant was informed in writing and orally about the aim of this study and gave informed consent.

## 3. Results

Four hundred and fifty-eight subjects participating in geriatric revitalization met the requirements of this study (age over 65 years, score ≥ 90 in the Barthel Index, and informed consent); 98 subjects (21.39%) were excluded at the end of the program because they lacked monitoring (Figure 1).

Regarding socio-demographic characteristics, 85.5% were women, and the mean age was 76.28 ± 5.90 years; 12% of the subjects were 65–69 years old, 59% were 70–79 years old, 27.5% were 80–89 years old, and 1.5% were older than 90. The majority lived with a companion or with someone who would care for them (only 20% live alone), 62% resided in the center of an urban area, and they have participated in the program for 5.48 ± 5.38 years. The number of conditions presented by the study participants ranged from 0 to 8, with a mean of 2.39 (SD 1.42). The Charlson comorbidity index, which predicts mortality at one year, was calculated, obtaining a result of low comorbidity for 25 of the elderly and the absence of comorbidity according to the index in the rest of the participants. Specifically, the presence of autoimmune diseases was analyzed, resulting in 69 participants (19.2%) suffering from some type of autoimmune disease, of whom 67 were women (97%). Frailty risk is available for comparison in Table 1 according to demographic and clinical features; statistically significant differences (*p* < 0.05) are found among older women living in urban areas, with a higher fat percentage and more pathologies and greater baseline medication use, plus lower scores in the Barthel Index and Lobo’s test. Moreover, there were no differences between physical activity (now and five years ago) nor time spent participating in the Geriatric Revitalization program.

In the baseline records, we identified heterogeneous results of subjects with high frailty risk: 44 participants (12.2%) with scores <10 points in SPPB, 27 participants (7.5%) with time records >12 s in TUG, and 14 participants (3.9%) with ≥3 positive answers to FRAIL scale questions; these data spread to 134 participants (37.8%) when taking scores ≥1 (Table 2) as frailty indicators. 

In Table 2, scores resulting from the three measurements are compared. SPPB shows that the percentage of robust patients changes from 47.2% to 62.2% after the intervention, mainly because of the change experienced by people originally considered frail. This difference can also be observed in mean scores (Week 1: 10.97 vs. Week 30: 11.22, *p* = 0.040) (Figure 2a). As for TUG, there is evidence that, even though improvement during the intervention is remarkable (Week 1: 8.72 vs. Week 30: 7.98, *p* < 0.001), the period without exercise has a reverse effect on the new situation (Week 30: 7.98 vs. Week 43: 8.45, *p* = 0.007) (Figure 2b). FRAIL, for its part, shows much more unspecific changes (Figure 2c) when it comes to considering cut points ≥1.

To conclude, Figure 3 describes the relationship between FRAIL (1a–1c) and SPPB (2a–2c) scores and TUG time records. The trend is validated by the analysis of correlation between tests, which shows a strong negative correlation between TUG and SPPB (Week 1 = −0.665, Week 30 = −0.731, Week 43 = −0.624; *p* < 0.001) and a weak or poor correlation between SPPB-FRAIL and TUG-FRAIL (<0.250).

## 4. Discussion

The data collected show that some specific socio-demographic and clinical characteristics can also relate to higher frailty risk values in trained populations. The various assessment tools in use in the study give heterogeneous results when analyzed considering the validated cut point for frailty during training periods and periods without training as well, but they are consistent throughout time as they point out improvement as a result of training. However, TUG shows a good correlation with SPPB if data are considered persistently.

An interesting aspect highlighted in this study is the identification of statistically significant differences among older women living in urban areas, particularly with a higher fat percentage, greater baseline medication use, and lower scores in functional assessments like the Barthel Index and Lobo’s test. These findings underscore the importance of considering socio-demographic factors and baseline health status when assessing frailty risk and designing intervention programs. Many authors refer to the influence of cofactors associated with frailty. Ménendez-González et al. [21] describe a woman’s profile: aged over 84 years old, with comorbidities and polypharmacy, whose data are comparable to our outcomes, although in their case frailty was assessed following Fried’s criteria. Ozkok et al. [22], for their part, did assess the role played by polymedication in subjects’ performance according to TUG and SPPB scales—a connection was found with the regression model, including low scoring in SPPB. It is important to remember that demographic and clinical factors are not the only ones to influence frailty; biopsychosocial aspects such as the quality of life and the dysfunctionality of the environment have also been related thereto [23].

Consequently, using these scales (SPPB and TUG) adds more specificity while evaluating the performance of other tools that focus on the subject’s day-to-day functional abilities or the ones the subject mentions (FRAIL), which allows full objectivity in the decision-making process and flexibility towards change [24]. The study’s baseline records reveal heterogeneous results in identifying high frailty risk, with varying percentages of participants exhibiting frailty indicators across different assessments (SPPB, TUG, and FRAIL). In compliance with the recommendations given in Spain with regards to diagnosing frailty in primary care [16], the SPPB scale and Walk Speed test are the most popular, while TUG and FRAIL remain as accepted alternatives. Protocolizing the circumstances under which each tool is employed is relevant, since screening varies a lot from one test to another [25]. The works by Pereiro et al. [26] and Río et al. [27] stand out as they give TUG and SPPB reference values for different cohorts. In our cohort of trained community-dwelling elderly, while SPPB and TUG demonstrate meaningful changes over the intervention period, the FRAIL scale shows more variable results, suggesting potential limitations in its sensitivity to detect changes in frailty status. This variability underscores the need for careful consideration of measurement tools and their suitability for assessing frailty in other populations.

Just as said, there is no agreement on the recommendations to be given about exercise to avoid frailty [14,15]. Previous research has demonstrated that multicomponent training in populations with heterogeneous baseline frailty status improves subjects’ results proportionally, so it is recommendable for both reversing frailty and avoiding it [28,29]. But according to Tangen and Robinson’s outcomes [30], several frailty assessment tests (SPPB and one-leg standing) should be dismissed in older people who exercise and have a high functional level because of the floor effect produced by their data. According to our results, the intervention’s effectiveness is demonstrated by improvements in functional assessments, particularly on the SPPB, where the percentage of robust patients increased significantly after the intervention period. However, there is a notable decline in functional improvements during periods of inactivity, highlighting the importance of consistent exercise for maintaining gains in frailty reduction.

This study has some limitations: primarily, it has been carried out considering a non-homogeneous sample regarding sex, age, and previous level of physical activity. Since the training program has already been implemented for a long time, it has been observed that the oldest people are the ones participating the longest in the program. One inclusion criterion of this study was scoring more than 90 points in the Barthel Index scale, which led to the exclusion of the most deteriorated subjects and set a limit for potential participants that showed frailty at the time of the baseline assessment. Finally, the concept of frailty is not clearly defined. Definitions aimed at the functional capacity of the older person are more objective than those based on the characteristics of the frail person. Even though there are a large number of frailty measurement instruments identified, and we opted for the SPPB, TUG, and FRAIL. The agreement between the three instruments for assessing frailty in our sample is weak despite being statistically significant. This lack of consensus on a single instrument for the diagnosis of frailty makes larger studies necessary with different cohorts.

## 5. Conclusions

Our study’s outcomes suggest that a 30-week training program based on multicomponent exercise improves frailty and pre-frailty results in a community-dwelling population with no functional decline. A lack of homogeneity is proven among the various tools used for measuring frailty probability over training periods and inactivity periods (SPPB, TUG, and FRAIL) when it comes to analyzing them by considering the validated cut point. However, TUG shows a good correlation with SPPB if data are considered persistently.

## Figures and Tables

**Figure 1 jcm-13-02514-f001:**
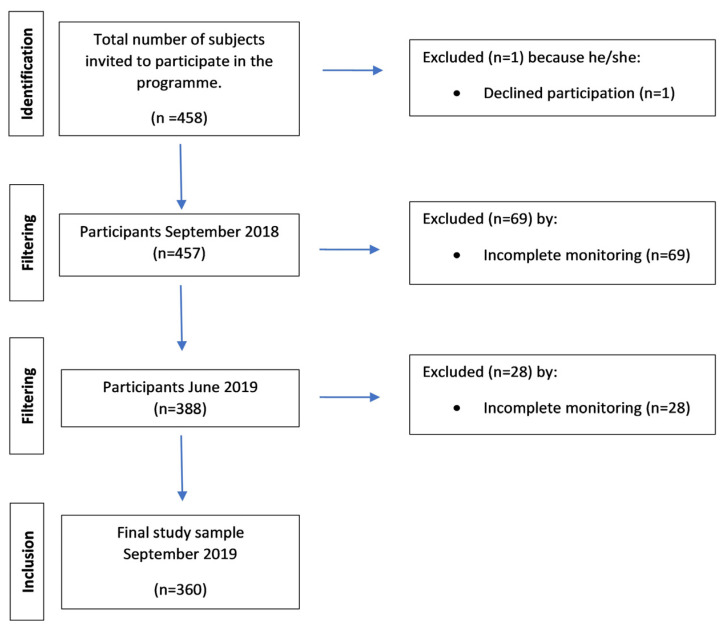
Flowchart.

**Figure 2 jcm-13-02514-f002:**
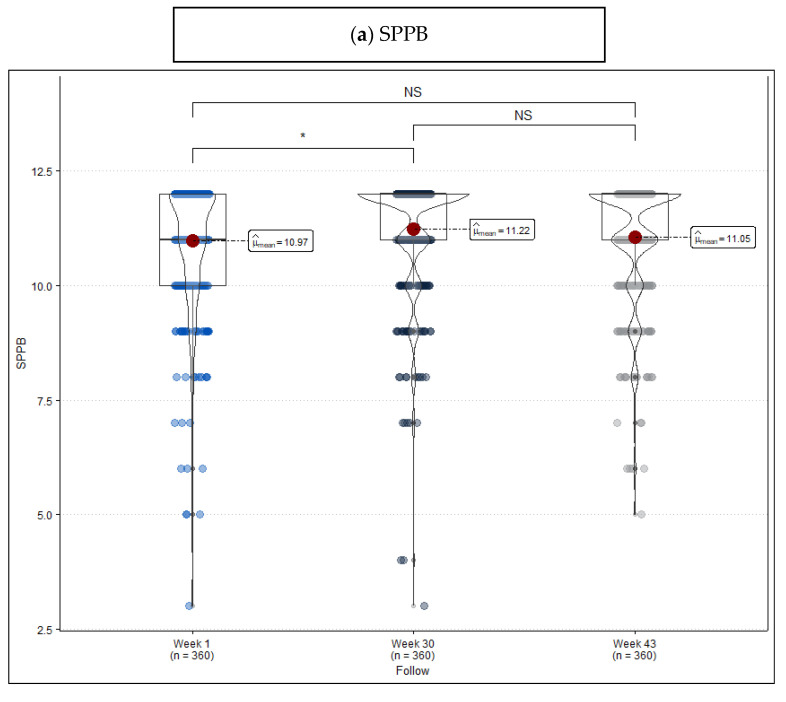
Violin plot frailty level score measured at the beginning of the program (week 1), at the end of the program (week 30), and after a period without exercise (week 43). (**a**) Short Physical Performance Battery (SPPB), (**b**) Timed “Up & Go” (TUG), and (**c**) FRAIL Questionnaire Screening Tool. Remarks: NS = no statistically significant differences, * *p*-value < 0.05, ** *p*-value < 0.01, and *** *p*-value < 0.001.

**Figure 3 jcm-13-02514-f003:**
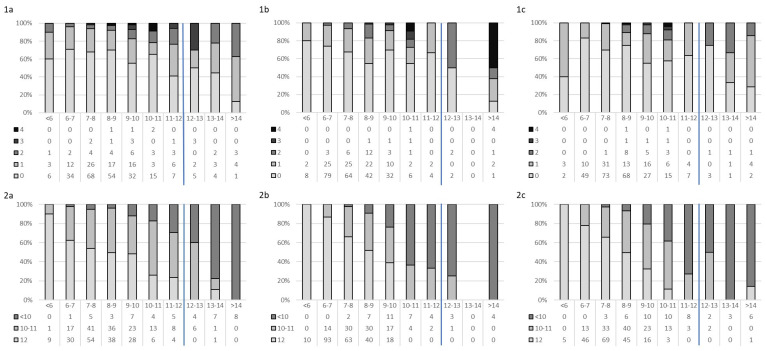
Distribution of the number of frailty components (1—FRAIL Questionnaire Screening Tool and 2—Short Physical Performance Battery (SPPB)) by Timed “Up & Go” (TUG). (**1a**,**2a**) Week 1, (**1b**,**2b**) Week 30, and (**1c**,**2c**) Week 43. Blue line for the cut-off.

**Table 1 jcm-13-02514-t001:** Baseline characterization by score on the Short Physical Performance Battery (SPPB), FRAIL Questionnaire Screening Tool, and Timed “Up & Go” (TUG).

	Baseline SPPB				Baseline FRAIL				Baseline TUG		
	12 Points (n = 170)	10–11 Points (n = 146)	<10 Points (n = 44)	*p*-Value	0 Items (n = 226)	1–2 Items (n = 120)	≥3 Items (n = 14)	*p*-Value	<12 s (n = 333)	≥12 s (n = 27)	*p*-Value
Age	75.38 ± 5.68	76.13 ± 5.77	80.25 ± 6.02	<0.001 ***	75.87 ± 5.84	76.65 ± 6.10	79.64 ± 5.15	0.049 *	76.02 ± 5.88	79.52 ± 5.88	0.003 **
Sex											
Male	28 (53.8%)	22 (42.3%)	2 (3.8%)	0.129	43 (82.7%)	9 (17.3%)	0 (0.0%)	0.004 **	52 (100.0%)	0 (0.0%)	0.026 *
Female	142 (46.1%)	124 (40.3%)	42 (13.6%)		183 (59.4%)	111 (36.0%)	14 (4.5%)		281 (91.2%)	27 (8.8%)	
Residence											
Urban	107 (47.6%)	88 (39.1%)	30 (13.3%)	0.628	134 (59.6%)	78 (34.7%)	13 (5.8%)	0.033 *	203 (90.2%)	22 (9.8%)	0.034 *
Peri-urban	63 (46.7%)	58 (43.0%)	14 (10.4%)		92 (68.1%)	42 (31.1%)	1 (0.7%)		130 (96.3%)	5 (3.7%)	
Cohabitation											
Alone	59 (45.7%)	52 (40.3%)	18 (14.0%)	0.001 **	80 (62.0%)	41 (31.8%)	8 (6.2%)	0.299	119 (92.2%)	10 (7.8%)	0.184
Married	97 (49.0%)	86 (43.4%)	15 (7.6%)		128 (64.6%)	66 (33.3%)	4 (2.0%)		186 (93.9%)	12 (6.1%)	
With children	14 (42.4%)	8 (24.2%)	11 (33.3%)		18 (54.5%)	13 (39.4%)	2 (6.1%)		28 (84.8%)	5 (15.2%)	
Previous Falls											
No	155 (46.5%)	136 (40.8%)	42 (12.6%)	0.585	211 (63.4%)	109 (32.7%)	13 (3.9%)	0.696	307 (92.7%)	26 (7.8%)	0.436
Yes	15 (55.6%)	10 (37.0%)	2 (7.4%)		15 (55.6%)	11 (40.7%)	1 (3.7%)		26 (96.3%)	1 (3.7%)	
Physical A. (5 years)											
None	37 (52.1%)	25 (35.2%)	9 (12.7%)	0.480	42 (59.2%)	26 (36.6%)	3 (4.2%)	0.655	63 (88.7%)	8 (11.3%)	0.268
Planned	110 (46.6%)	95 (40.3%)	31 (13.1%)		151 (64.0%)	74 (31.4%)	11 (4.7%)		218 (92.4%)	18 (7.6%)	
Walking	16 (38.1%)	22 (52.4%)	4 (9.5%)		25 (59.5%)	17 (40.5%)	0 (0.0%)		41 (97.6%)	1 (2.4%)	
Balance	7 (63.6%)	4 (36.4%)	0 (0.0%)		8 (72.7%)	3 (27.3%)	0 (0.0%)		11 (100.0%)	0 (0.0%)	
Physical A. (present)											
None	19 (48.7%)	14 (35.9%)	6 (15.4%)	0.752	20 (51.3%)	16 (41.0%)	3 (7.7%)	0.634	34 (87.2%)	5 (12.8%)	0.554
Planned	144 (48.0%)	121 (40.3%)	35 (11.7%)		191 (63.7%)	98 (32.7%)	11 (3.7%)		279 (93.0%)	21 (7.0%)	
Walking	5 (29.4%)	9 (52.9%)	3 (17.6%)		12 (70.6%)	5 (29.4%)	0 (0.0%)		16 (94.1%)	1 (5.9%)	
Balance	2 (50.0%)	2 (50.0%)	0 (0.0%)		3 (75.0%)	1 (25.0%)	0 (0.0%)		4 (100.0%)	0 (0.0%)	
Weight	66.69 ± 10.85	67.06 ± 10.84	67.24 ± 11.39	0.933	67.01 ± 10.46	67.24 ± 11.45	62.46 ± 12.41	0.292	66.91 ± 10.98	66.84 ± 9.87	0.973
BMI	28.53 ± 4.42	28.54 ± 3.97	29.20 ± 4.36	0.623	28.23 ± 3.72	29.33 ± 4.91	28.74 ± 5.13	0.070	28.54 ± 4.21	29.56 ± 4.38	0.228
Fat %	42.10 ± 5.33	42.84 ± 4.66	44.31 ± 4.61	0.043 *	42.11 ± 5.18	43.22 ± 4.58	46.90 ± 2.82	0.001 **	42.45 ± 5.08	45.49 ± 2.83	<0.001 ***
Years in program	5.20 ± 5.02	5.26 ± 5.50	7.30 ± 6.10	0.058	5.23 ± 5.21	5.82 ± 5.70	6.71 ± 5.44	0.427	5.43 ± 5.36	6.11 ± 5.70	0.528
No. of conditions	2.26 ± 1.50	2.38 ± 1.47	2.91 ± 2.50	0.038 *	2.09 ± 1.23	2.72 ± 1.71	4.43 ± 1.15	<0.001 ***	2.38 ± 1.48	2.56 ± 1.64	0.547
No. of medicines	2.96 ± 2.18	2.99 ± 2.14	4.16 ± 2.50	0.004 **	2.72 ± 1.95	3.51 ± 2.35	6.14 ± 2.68	<0.001 ***	3.08 ± 2.22	3.56 ± 2.43	0.290
Barthel Index	93.85 ± 2.10	93.80 ± 2.14	92.50 ± 2.52	0.001 **	94.12 ± 1.91	93.00 ± 2.46	92.14 ± 2.56	<0.001 ***	93.71 ± 2.19	93.15 ± 2.46	0.260
Lawton Brody	7.96 ± 0.29	7.92 ± 0.43	7.86 ± 0.34	0.259	7.93 ± 0.34	7.98 ± 0.15	7.57 ± 1.08	<0.001 ***	7.93 ± 0.37	7.93 ± 0.26	0.945
Yesavage	2.24 ± 2.38	2.14 ± 2.09	2.70 ± 2.18	0.345	1.79 ± 1.95	2.92 ± 2.45	4.07 ± 2.58	<0.001 ***	2.21 ± 2.27	2.78 ± 1.78	0.210
Lobo	29.26 ± 4.18	29.82 ± 4.36	28.32 ± 4.30	0.112	29.75 ± 4.15	28.92 ± 4.32	27.21 ± 5.38	0.036 *	29.54 ± 4.22	27.33 ± 4.56	0.010 *

* *p*-value < 0.05, ** *p*-value < 0.01, and *** *p*-value < 0.001; age: <10 vs. 10–11 = <0.001, <10 vs. 12 = <0.001, fat %: <10 vs. 12 = 0.044, no. of conditions: <10 vs. 12 = 0.032, no. of medicines: <10 vs. 10–11 = 0.007, <10 vs. 12 = 0.004, Barthel Index: <10 vs. 10–11 = <0.002, <10 vs. 12 = <0.001; fat %: 0 vs. ≥3 = 0.001, 1–2 vs. ≥3 = 0.028, No. of conditions: 0 vs. 1–2 = <0.001, 0 vs. ≥3 = <0.001, 1–2 vs. ≥3 = <0.001, no. of medicines: 0 vs. 1–2 = 0.003, 0 vs. ≥3 = <0.001, 1–2 vs. ≥3 = <0.001, Barthel Index: 0 vs. 1–2 = 0.001, 0 vs. ≥3 = 0.003, Lawton Brody: 0 vs. ≥3 = 0.001, 1–2 vs. ≥3 = <0.001, Yesavage: 0 vs. 1–2 = <0.001, 0 vs. ≥3 = <0.001.

**Table 2 jcm-13-02514-t002:** Number of subjects compared by score on the Short Physical Performance Battery (SPPB), FRAIL Questionnaire Screening Tool, and Timed “Up & Go” (TUG) one measuring at a time (week 1, week 30, and week 43).

		Week 1	Week 30	Week 43	*p*-Value
SPPB	12 points	170 (47.2%)	224 (62.2%)	185 (51.4%)	
	10–11 points	146 (40.6%)	98 (27.2%)	127 (35.3%)	
	<10 points	44 (12.2%)	38 (10.6%)	48 (13.3%)	
	±DE Average	10.98 ± 1.39	11.23 ± 1.34	11.05 ± 1.32	0.040 *
FRAIL	0 items	226 (62.8%)	238 (66.1%)	247 (68.6%)	
	1–2 items	120 (33.3%)	118 (32.8%)	108 (30.0%)	
	≥3 items	14 (3.9%)	4 (1.1%)	5 (1.4%)	
	±DE Average	0.54 ± 0.84	0.44 ± 0.69	0.40 ± 0.68	0.041 *
TUG	<12 s	333 (92.5%)	352 (97.8%)	346 (96.1%)	
	≥12 s	27 (7.5%)	8 (2.2%)	14 (3.9%)	
	±DE Average	8.71 ± 2.27	7.98 ± 2.04	8.44 ± 1.84	<0.001 ***

*p*-value: * < 0.05/** < 0.01/*** < 0.001; post-hoc Bonferroni: SPPB week 1 vs. week 30 = 0.040 */FRAIL week 1 vs. week 43 = 0.043 * TUG week 1 vs. week 30 < 0.001 ***/TUG week 30 vs. week 43 = 0.007 **.

## Data Availability

All necessary data are supplied and available in the manuscript; however, the corresponding author will provide the dataset upon request.

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
