# Peer review of "A Prospective Observational Study of Frailty in Geriatric Revitalization Aimed at Community-Dwelling Elderly"

_jcm, 2024, doi:10.3390/jcm13092514_

Round 1

Reviewer 1 Report

Comments and Suggestions for Authors

Comments on the Quality of English Language

The text needs minor English editing.

Author Response

Dear reviewer,

We wish to thank you for your constructive comments in this review. Your comments have provided valuable insights to help us refine the content and its analysis.  We try to address the issues raised as best as possible in the document attached.

Best regard,

The authors.

Reviewer 2 Report

Comments and Suggestions for Authors

The manuscript "A prospective observational study of frailty in geriatric  revitalization aimed at community-dwelling elderly" concerns a very crucial health problem of frailty syndrome that can be prevented or even reversed using the proper and systematically applied procedures. Besides a high-protein diet, a multitask training is a very efficient tool to approach the better level of physical and functional status as well better  quality of life in elderly population.

A prospective observational study of a multicomponent exercise program for geriatric revitalization described in this paper contains a well-recruited and large enough group of 360 persons aged over 65 years in whom frailty levels were assessed with usage of the standardized tools.  The program was developed long enough (over 30 weeks), with the proper intensity and frequency (three times a week, in sessions lasting 45-50 minutes each). The interesting results are presented in the clear and extensive manner.

The only remark: could the authors mention about the most commonly identified  clinical conditions in the examined group and a percentage of multimorbidity cases?

Author Response

(The authors gave the same response as above.)

Reviewer 3 Report

Comments and Suggestions for Authors

Here the authors examine an exercise program that started in 1994. 

I see no scientific novelty here. Yes. The authors did a lot of stastics, but I do not see the novelty of this work.  

Author Response

(The authors gave the same response as above.)

Round 2

Reviewer 1 Report

Comments and Suggestions for Authors

The authors have taken into account all the comments of the reviewer, the paper is suitable for publication in present form.